# What shapes feature representations?
# Exploring datasets, architectures, and training

**Katherine L. Hermann**[*]
Stanford University
hermannk@stanford.edu

**Andrew K. Lampinen**[*]
Stanford University
andrewlampinen@gmail.com

## Abstract

In naturalistic learning problems, a model's input contains a wide range of features, some useful for the task at hand, and others not. Of the useful features, which ones does the model use? Of the task-irrelevant features, which ones does the model represent? Answers to these questions are important for understanding the basis of models' decisions, as well as for building models that learn versatile, adaptable representations useful beyond the original training task. We study these questions using synthetic datasets in which the task-relevance of input features can be controlled directly. We find that when two features redundantly predict the labels, the model preferentially represents one, and its preference reflects what was most linearly decodable from the untrained model. Over training, task-relevant features are enhanced, and task-irrelevant features are partially suppressed. Interestingly, in some cases, an easier, weakly predictive feature can suppress a more strongly predictive, but more difficult one. Additionally, models trained to recognize both easy and hard features learn representations most similar to models that use only the easy feature. Further, easy features lead to more consistent representations across model runs than do hard features. Finally, models have greater representational similarity to an untrained model than to models trained on a different task. Our results highlight the complex processes that determine which features a model represents.

## 1 Introduction

How does a deep neural model see the world at initialization, and what changes over the course of training? If there are many latent features in the training data — such as the colors, textures, and shapes of objects in visual datasets — how does the model separate task-relevant features from irrelevant ones? Does a model gain sensitivity to diagnostic features by enhancing its representation of them, or by suppressing its representations of other features? What does a model represent among task-irrelevant features, features that are task-relevant but unreliable, and features that are redundantly predictive, each of which are present in naturalistic datasets? How similar are feature representations across models as a function of training task? How does the similarity depend on the feature?

From an engineering perspective, these questions are important as the field tries to build models whose representations are versatile and general-purpose enough to support out-of-distribution generalization, and to transfer to new downstream tasks. For example, it is common practice to transfer weights or activations from ImageNet models to models used for other vision tasks [15, e.g.], and to use BERT [7] as a starting point for language tasks. Yet untrained models can sometimes provide an initialization that is nearly as good as ImageNet pretraining [12, 23], raising the questions of what information is present in an untrained model, and how it is modified during training. Furthermore, models sometimes learn to use "shortcut features" [10] – an object's texture, rather than shape

---

[*]Contributed equally

[11], for example – that solve the training task, but fail to generalize robustly. This phenomenon highlights the importance of understanding which features are learned in a given setting, especially when features are correlated. Additionally, understanding how models select features is important for ensuring that models make equitable and non-biased decisions [45, 47].

From a scientific perspective, understanding which features a model learns, and how consistent feature representations are across model instances, is relevant to understanding how models compute [2], as well as how they should be compared to one another [29] or to neural data [19, 6, 5, 39, 43].

In this paper, we investigate the evolution of models' feature preferences using synthetic datasets in which we can directly control the relationships between multiple features and task labels, which is not possible in naturalistic datasets like ImageNet. We create datasets from a data-generating process in which underlying latent variables give rise to input features like shape or texture (in our visual tasks), or linearly or non-linearly extractable features (in our binary features tasks). We train models on tasks that are either deterministically or probabilistically related to these input features. Leveraging the complementary approaches of decoding and representational similarity analysis [24], we probe feature representations within and between models across datasets, architectures, and training. Our results provide insight into how feature representations are affected by learning and how to analyze them. Our key contributions are:

- We find that many input features are already partially represented in the higher layers of untrained models. Training on a task enhances task-relevant features (increases their decodability relative to an untrained model), and suppresses both task-irrelevant features *and* some task-relevant features.

- We investigate what a model represents when multiple features predict the label. We find that, when a pair of features redundantly predicts the label, models prefer one of the features over the other, and the preference structure tracks untrained decodability. When only one feature is perfectly predictive, we find that the representations of a correlated feature remain roughly constant as a function of correlation, until the correlation becomes quite high.

- We identify cases in which models are "lazy," and suppress a more predictive feature in favor of an easier-to-extract, but less predictive, feature.

- We find that easy features induce representations that are more consistent across model runs than do hard features, and that a multi-task model trained to report both easy and hard features produces representations that are very similar to those of a model trained only on the easy feature.

## 2 Related Work

Both theoretical and empirical work has investigated how models represent task-irrelevant features over training. Theoretical work by Shwartz and colleagues [38] suggests that deep models compress away task-irrelevant information as they learn a task, though the generality of this observation has been disputed [34]. Several papers have [14, 16] shown that it is possible to decode "category-orthogonal" features from CNNs trained on naturalistic datasets. However, catastrophic interference between new tasks and prior tasks shows that at least some features are suppressed or altered during training [28, 21, 46, 33, 27]. Thus, a tension exists in the literature about how features change over learning: is task-irrelevant information suppressed, or just ignored?

Theoretical work by Saxe and colleagues has found that supervised linear models learn the input features that account for the most variance in the data before learning features that account for less variance [35]. Given hierarchical data, models learn high-level semantic features before proceeding to lower-level ones [36]. This observation that models progressively differentiate task structure has been leveraged to explain the generalization benefits of optimal stopping: since stronger features have a higher signal-to-noise ratio, optimal stopping captures features to the extent that they are both important and not overly noisy [25]. Qualitatively connecting with this work, we ask the further question of how feature difficulty interacts with feature strength. For example, are features that are linearly extractable learned before features that require nonlinear extraction? How does a model trade off ease of extracting a feature and that feature's reliability in predicting task labels?

Some work suggests that deep networks may have an inherent bias to favor simple functions. For example, there exist many more parameter choices which yield simple than complex functions [41]. The inductive biases of SGD in highly over-parameterized models might contribute to finding solutions that meet various notions of simplicity [3, 4]. Our empirical results help ground these

theoretical predictions, and connect to prior explorations showing that deep networks sometimes exploit "shortcut features" – features which may be sufficient to solve a training task, but which may fail to generalize robustly and differ from the features preferred by people [10]. For example, in the vision domain, recent work has found that standard ImageNet models prefer to classify objects by texture rather than shape, whereas the reverse is true for people [11], though this bias can be ameliorated with data augmentation [11, 13]. Whereas Geirhos and colleagues studied the classification (output) behavior of models trained on ImageNet, a dataset whose joint image feature–label statistics are unknown and uncontrolled, our experiments investigate the decodability of visual features from intermediate model layers when trained on datasets with controlled statistics. Using analyses more similar to ours, [13] found that shape information is more decodable than texture in the convolutional layers of an ImageNet-trained CNN. Concurrent work by Shah and colleagues observes that models may prefer simple (e.g. linear) features even when difficult (e.g. non-linear) ones are more reliable [37], and Parascandolo et al. propose a new approach to optimization that could potentially ameliorate this affect [31].

Related to the question of which features models come to represent over training is the question of which features the model represents at initialization. Prior work has observed that the representations in untrained networks already transfer well to tasks like classification [17, 32], and denoising and inpainting ([40]). Evolved architectures with untrained weights have been successful in action-planning problems [9]. Further work has shown that minimal training – only adjusting BatchNorm parameters – of untrained models already yields surprisingly high classification performance [8]. In addition, untrained initializations often perform surprisingly similarly to pretrained ones after fine-tuning on a new task [12, 23]. Furthermore, features in untrained models can explain substantial amounts of neural data, although trained models generally perform slightly better [44, 18, 39, 6, 5]. Overall, this prior work observes that untrained feature representations capture a great deal of task-relevant structure; we explore how these initial representations change over training.

## 3    Does feature selection happen by enhancement or suppression?

Over training, as a model becomes sensitive to a target feature, does it build this sensitivity by enhancing the target feature, or by suppressing non-target features? How much enhancement and/or suppression occurs? In this set of experiments, we used two synthetic datasets, and trained vision models to classify images according to their shape, texture, or color. We then tested to what extent target and non-target features were decodable before and after training.

**Datasets**. In a set of preliminary experiments, our datasets consisted of $224 \times 224$ RGB images of objects with multiple features, each of which could be used as a label in a classification task. For each of these tasks, we created 5 cross-validation splits, creating validation sets by holding out a subset of values for the non-target features (3 classes per feature). For example, a validation set for a color classification task might include squares, triangles, and circles, while no training examples would have had these shapes. Thus, our classification tasks required generalizing the target feature (e.g. color) over the non-target feature(s) (e.g. shape and texture).

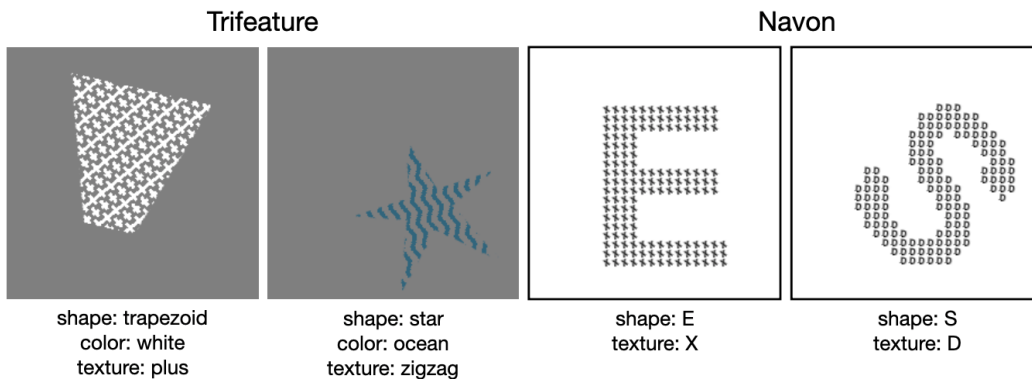

Figure 1: Example vision dataset items. The Navon dataset is adapted from [13] and based on [30].

***Navon dataset***. As shown in Figure 1, this dual-feature dataset, modified from [13] based on [30], consisted of images in which a large letter ("shape") is rendered in small copies of some other letter ("texture"). We rendered each shape-texture combination ($26\times26$) at 5 positions, rotating the shape and texture independently at angles drawn from $[-45, 45]$ degrees, yielding 3250 items after excluding those with the same shape and texture.

***Trifeature dataset***. Images contained one of ten shapes, rendered in one of ten textures, in one of ten colors (see Figure B.1). Shapes were rendered within a $128 \times 128$ square, which was rotated at an angle drawn from $[-45, 45]$ degrees and placed at a random position within the larger image, such that the shape was completely contained within the image, before an independently rotated texture and a color were applied. Textures were rendered so that they could be discriminated within at most a $10 \times 10$ square region. We rendered 100 versions of each color, shape, and texture combination, for a total pool of 100,000 images. For the experiments in this section, we created train sets of 3430 items and validation sets of 3570 images, in both of which the features were uncorrelated.

**Model architecture**. We evaluated two model architectures: AlexNet and ResNet50 with output sizes modified to reflect the number of target classes (Navon: 23, Trifeature: 7). AlexNet's other fully connected layers were also narrowed proportionally (Navon: 95 units each in fc6, fc7; Trifeature: 29 units each). An interesting question for future follow up is the effect of fully-connected layer width on feature representations. We used the `torchvision` implementation of the both models' convolutional layers [42], and fixed weight initializations across experiments.

**Training procedure**. Training and validation images were normalized by the mean and standard deviation of the training data. They were not cropped. All models were trained for 90 epochs using Adam optimization [20] with a learning rate of $3 \times 10^{-4}$, weight decay of $10^{-4}$, and batch size of 64, using a modified version of the `torchvision` training script [1]. We selected the model corresponding to the highest validation accuracy over the training period.

**Decoding**. To determine which visual features a trained model *represents* – that is, which features (of shape, color, and texture) can be extracted from layer activations of the trained model in response to a set of images – we trained decoders to map activations from some layer of a trained, frozen model to feature labels using a single linear layer followed by a softmax (multinomial logistic regression). Importantly, the labels on which the decoder was trained were not, in the general case, the same as the labels with which the network was trained. We decoded from upper layers of models (AlexNet: pool3, fc6, or fc7; ResNet-50: pre-pool, post-pool). See Appendix B.1.1 for additional details.

### 3.1 What's already decodable from an untrained model?

Across two datasets (Navon and Trifeature) and two model architectures (AlexNet and ResNet-50), visual features were decodable significantly above chance from the upper layers of untrained models. As shown in Figures 2 and A.2, the rank order of decoding accuracy by feature held across architectures and across the layers of AlexNet: color was most decodable, followed by shape and then texture. Color was perfectly decodable from the final convolutional layer of AlexNet as well as from ResNet-50. Interestingly, decoding accuracy for all features decreased through the fully connected layers of AlexNet.

In the Navon dataset (Figure 3), it was possible to decode shape and texture with accuracy >60% (chance = 4.3%) from the final convolutional layer of AlexNet. Consistent with the Trifeature results, decoding accuracy fell off through the fully-connected layers, but was still above chance in the penultimate layer. Both features were decodable above chance from ResNet-50, at appx. 25% prior to the global average pool, dropping off slightly after the pool (see Figure A.1).

One hypothesis is that the decodability of features from an untrained model reflects the model's inductive biases, and might predict the extent to which a feature would be preserved after training the model on a different task.

### 3.2 What's decodable after training?

In this section, we trained models to classify a *target feature* in the presence of one or more *non-target*, task-irrelevant (uncorrelated with the label) features.

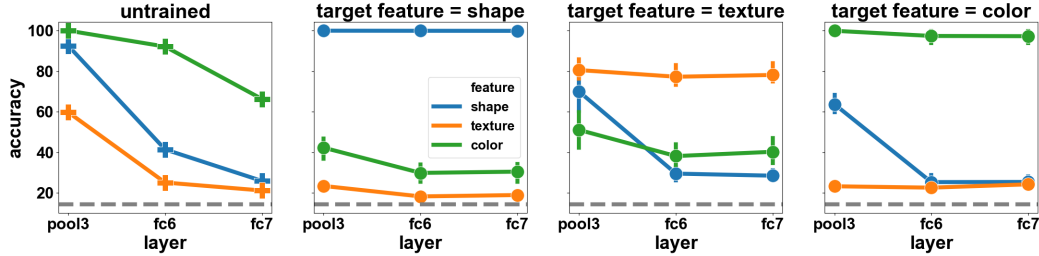

Figure 2: **Target features are enhanced, and non-target features are suppressed, relative to an untrained model in models with an AlexNet architecture trained on Trifeature tasks.** Accuracy decoding features (shape, texture, color) from layers of an untrained model (far left) versus from models trained to classify shape, texture, or color (mean decoding accuracy across models trained on each of 5 cv-splits of the data; error bars indicate 95% CIs). Chance = $\frac{1}{7}$ = 14.3% (dashed gray line). Decoding accuracy is generally higher for target features in the trained than in the untrained model (enhanced) and lower for non-target features (suppressed).

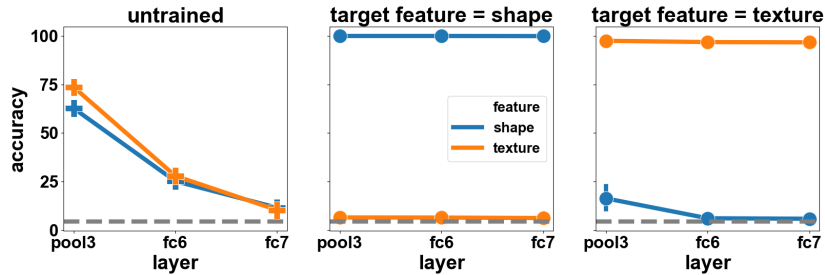

Figure 3: **Feature decodability in models with an AlexNet architecture trained on the Navon dataset.** Accuracy decoding features (shape, texture) from an untrained model (left) versus from shape- (center) and texture-trained models (right). Results corresponding to trained models are mean across models trained on 5 cv splits. Chance = $\frac{1}{23}$ = 4.3% (dashed gray line). As for the Trifeature models (Figure 2), decoding accuracy in the untrained model decreases across layers, and the target features are enhanced, whereas the non-target features are suppressed.

Across architectures (AlexNet and ResNet-50), target features were ***enhanced***: decoding accuracies were higher than in the untrained model, if they were not already at ceiling (Figures 3, 2, A.1, and A.2). Target features were highly decodable from all layers. By contrast, non-target features were partially ***suppressed***: decoding accuracies were lower than in the untrained model, but still above chance. Non-target features were most decodable from the the final convolutional layer of AlexNet and their decodability decreased layerwise, recapitulating the pattern found in the untrained models.

## 4    What if multiple features are predictive?

So far, we have considered which features a model represents when trained on datasets in which one feature is task-relevant and the others are not. However, in many real-world datasets, features exhibit some degree of correlation with one another. For example, in real-world objects, features like shape, texture, and color are often correlated, especially for natural objects. Here, we tested which features a model represents when multiple features predict the target to varying degrees.

**Datasets**. In these experiments, we used two datasets that allowed us to control the predictivity of features as well as feature difficulty.

*Trifeature (Correlated) datasets*. We created versions of the Trifeature dataset with correlated features: we sampled train sets (4900 images) and validation sets ($\geq$ 4900 images, varied with correlation) from the larger set of 100,000 images. We correlated a pair of features (e.g. shape and color) by choosing a conditional probability of one feature matching the other between 0.1 and 1. If this probability was 1, the features were perfectly correlated, and if it was 0.1, the features were uncorrelated. A pair of features was correlated across the set of images (not within individual images).

As an example, suppose that shape and color are correlated with conditional probability = 0.5. Then, in images containing triangles, half would be red and half would be a color sampled uniformly at random from the other color options (blue, white, etc.). Similarly, in images containing trapezoids, half would be orange and half would be some other color. The attribute matching (e.g. triangle = red, trapezoid = orange, etc.) was assigned randomly.

In each dataset version, training data contained 7 classes per input feature type (shape, color, texture), with 3 classes each held out. As in the uncorrelated trifeature datasets (above), validation set stimuli always had at least one of the non-target features held-out; for example, if the target feature was texture, generalization would be evaluated on shapes and colors that were not encountered during training. Feature correlations were matched in the validation set.[2]

***Binary Features datasets***. We created (non-vision) datasets containing features that differed in difficulty. We define a feature's difficulty in terms of the minimum complexity of a network required to extract it. We considered two features: an "easy"/"linear" one extractable by a single linear layer, and a "difficult"/"nonlinear" one requiring an XOR computation which must be implemented by a multi-layer perceptron [26]. The inputs in our dataset were 32-element binary (1 or 0) vectors in which the first 16 elements instantiated the easy/linear feature, and the second 16 instantiated difficult/nonlinear feature. The label was probabilistically determined by these two features — that is, the inputs were sampled so that each feature matched the label a certain percentage of the time. Specifically, we varied the predictivity of the easy feature ($p$ (label|easy feature)) between 0.5 (chance) and 1 (perfectly predictive). We fixed the predictivity of the harder feature at 0.9. We trained on 256 examples, and used 512 for evaluation and decoding.

## 4.1 What if two features redundantly predict the label?

Using the Trifeature (Correlated) datasets with conditional probabilities of 1.0, and methods as in Section 3.2, we trained models on classification tasks in which pairs of features (shape and color, or shape and texture) redundantly predicted the label, while the third feature was task-irrelevant. We then tested decodability of the predictive features (see Appendix B.1.1).

**Results**. Across architectures, rather than representing both features equally, one of the two predictive features was always substantially more decodable from the trained model than was the other. Intriguingly, the less decodable, but still perfectly predictive, feature was sometimes even suppressed relative to an untrained model. In AlexNet (Figure 4), when shape and color were

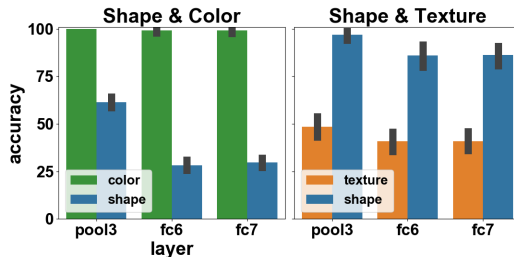

Figure 4: **When two features redundantly predict the label, models preferentially learn one feature.** Color is more decodable than shape, and shape is more decodable than texture, when AlexNet is trained on perfectly predictive pairs.

correlated, color was nearly perfectly decodable, whereas shape was considerably less so, and was in fact less decodable from pool3 and fc6 than in an untrained model (Figure A.7). When shape and texture were correlated, decodability of shape was considerably higher than decodability of texture, and texture was less decodable from pool3 than in an untrained model. The same rank-order decodability patterns held in ResNet-50 (Figure A.6, A.8). Overall, these results suggest that, when two features redundantly predict the label, rather than learning both, models privilege color over shape, and shape over texture. This preference structure aligns with the rank-order decodability of features from untrained models (Figures 2 and A.2).

## 4.2 What if one feature perfectly predicts the label, but another only partially predicts it?

In these experiments, in which we used Trifeature (Correlated) datasets with conditional probabilities of 0.1 to 0.9[3], one feature perfectly predicted the label, and a second feature (*correlated non-target*)

was correlated with the target feature to varying degrees across datasets. The third feature was uncorrelated with the others. We hypothesized that the decodability of the correlated non-target feature would increase as a function of its correlation with the target feature, possibly becoming enhanced relative to the untrained model.

**Results**. The target feature was always enhanced (higher decodability from the trained than the untrained model), whereas the correlated non-target feature was generally preserved (no change) or suppressed (Figure 5). Surprisingly, the level of suppression was more or less constant across degree of correlation, except at high correlations. When the conditional probability was 0.9, the correlated non-target features were slightly enhanced. A similar pattern was observed in the post-pool layer of ResNet-50 (Figure A.3). Figures A.4 and A.5 show raw decoding accuracies.

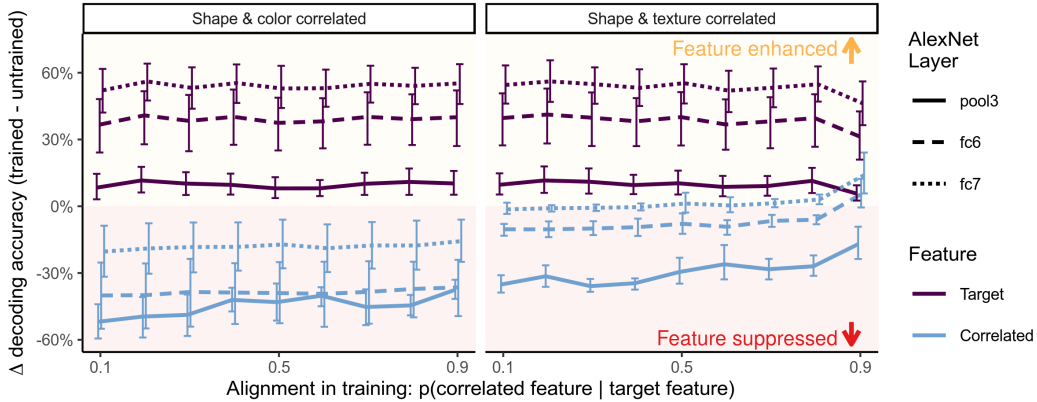

Figure 5: **Non-target features correlated with the target feature are generally suppressed.** For both datasets in which shape and color (left), and shape and texture (right), are correlated, non-target features correlated with the target feature ("correlated") are suppressed (red region) relative to the untrained model, whereas target features are enhanced (yellow region). Suppression of the correlated non-target feature is largely constant across correlation strengths (x axis), until very high correlations.

### 4.3 Do models prefer reliable but difficult features, or easy but less reliable ones?

Above, we tested what vision models learn when one feature perfectly predicts the label. Here, we consider the non-visual Binary Features datasets and a simpler model architecture (a 5-layer MLP), and consider what happens when no feature is perfectly predictive. We vary both the difficulty and predictivity of features, and test whether a model prefers a feature that is easier to learn (linearly decodable from the input) but only moderately predictive of the label, or a feature that is more difficult (XOR/parity, not linearly correlated with the input) but more predictive of the label. Will a model trade off predictivity for learnability?

**Models**. We used 5-layer MLPs (layer sizes 256, 128, 64, and 64). Hidden-layer nonlinearities were leaky rectifiers; output was a sigmoid. Training was via a cross-entropy loss, by full batch gradient descent (lr = 0.001). Decoding and representation analyses were performed at the final hidden layer (see Appendix B.1.2). We assessed feature reliance by testing on a dataset where the other feature was made unpredictive.

**Results**. The more reliable feature can be suppressed by a less reliable, but easier, one (Figure 6). When the easy feature had low predictivity, the model relied on the difficult feature (Figure 6, left panel). As the easier feature became more predictive, the model relied on it more, and less on the difficult feature. This switch happened before the predictivity of the features was matched; for example, the model preferred the easy feature when that feature's predictivity was 0.8, despite the harder feature having greater predictivity (0.9). Again, this preference for the easier feature reflects its greater decodability from an untrained model (see Figure A.9). Thus, in this case the model appears to trade off predictivity for ease of learning.

Which features are represented? The easy feature could be decoded with relatively high accuracy even if it was not predictive, but the difficult feature could be decoded reliably only when the feature-specific tests showed it was being used. (Even non-linear decoders could not recover the difficult feature much more accurately, see Figure A.10.) Intriguingly, a single input unit from either set of

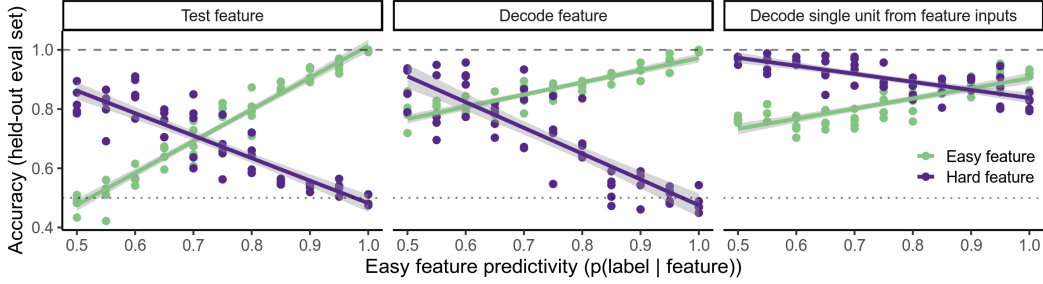

Figure 6: **More reliable, but difficult, features can be suppressed by less reliable, but easier, features.** The difficult feature (purple) had fixed predictivity of 0.9, while the easy feature (green) had varied predictivity (x-axis). The left panel evaluates the model's use of each feature (using a test set with the other feature made unpredictive). The middle panel shows decodability of each feature from the penultimate layer. The right panel shows decodability of a single input unit's value (another linear feature) from the inputs associated with each feature. (5 runs per condition; lines are linear fits.)

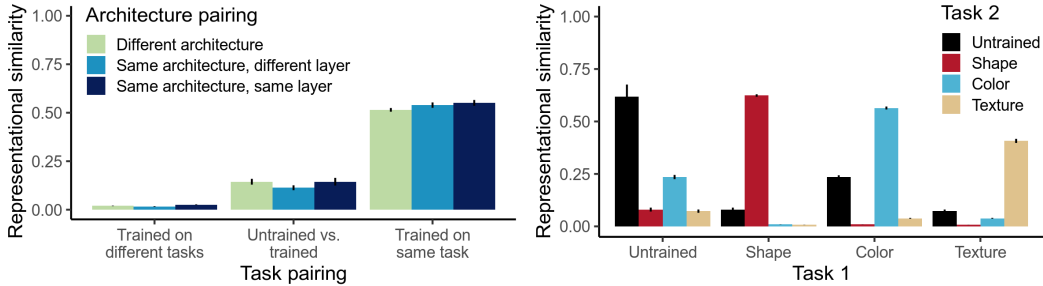

(a) Similarity by architecture and task pairing.  (b) Details of cross-task similarity.

Figure 7: **Representational similarity of Trifeature (Uncorrelated) models.** (a) Representational similarity of the same layer across two runs, of different layers within an architecture (e.g. pool3 and fc6), and of different layers across architectures (AlexNet pool3 and ResNet-50 post-pool). (b) Representational similarity across layers and architectures on each possible pair of tasks. Similarity is highest between pairs of models trained on the same versus different tasks, though within tasks, it is lowest for texture. (Results from 5 runs per condition; errorbars are bootstrap 95%-CIs.)

inputs could be decoded with around $80\%$ accuracy even if the associated feature was not predictive (Figure 6, right panel), showing that irrelevant inputs were not completely suppressed. However, complex, nonlinear combinations of those input units were difficult to recover.

## 5   What affects the representational similarity between models?

Our findings above suggest that which features a model represents depends on both the predictivity of features, and their availability in an untrained model/easiness to learn. In the following experiments, we tested whether models trained on the same task have consistent representations of inputs, and how similar their representations are to those of untrained models and models trained on a different task. To quantify similarity, we used Representational Similarity Analysis (RSA) [24], a method widely used in neuroscience, e.g. to compare models to neural (e.g. fMRI and physiology) data.

**Representation analyses**. For each model layer, for each training regime, we constructed a representational dissimilarity matrix (RDM), $\mathbf{D}^{S \times S}$, where $S$ is the number of stimuli (dataset inputs), and each entry $\mathbf{D}_{ij}$ contains a measure of the dissimilarity of patterns of activations elicited by stimuli $i$ and $j$. We used correlation distance as our dissimilarity measure. We then tested the level of correspondence between pairs of model layers by computing a Pearson's correlation of the upper right triangles of their RDMs. See Appendix B.4 for results with intermediate easy-feature predictivities, as well as with other analyses and metrics.

**Visual tasks**. In Figure 7 we show RSA results for the uncorrelated Trifeature tasks (see Figure A.11 for the Navon tasks). Models trained on matching tasks had much more similar representations than models trained on different tasks; in fact, models trained on one task were more similar to *untrained* models than to models trained on a different task. The magnitude of the similarity varied substantially, however. Similarity between two models trained on the (more difficult) texture task was lower (mean 0.408, bootstrap 95%-CI [0.394, 0.420]) than similarity between two models trained on one of the other tasks (e.g. shape, 0.624 [0.615, 0.632]), or even two untrained models (0.619 [0.542, 0.702]). The influence of architecture or layer on similarity was small compared to the influence of task. Similarity was most sensitive to the target feature up to relatively high correlations of other features (Figure A.12).

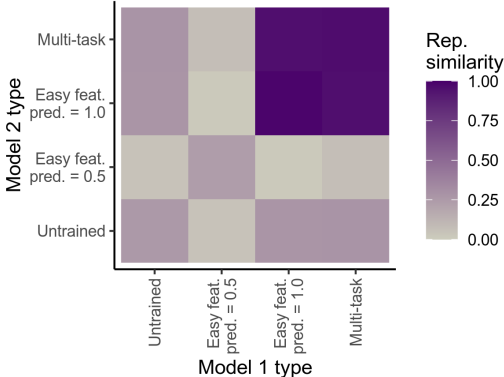

**Binary tasks**. Figure 8 contains RSA results in the binary features domain (see also Figure A.13). The similarity between two models trained with a highly predictive easy feature is much greater than the similarity between two models trained with an unpredictive easy feature. That is, models seem to find more representationally variable solutions to extracting a nonlinear feature than extracting a linear feature. In Figure A.14 we show some intuitions for why this might be the case. Furthermore, the untrained model and multi-task trained model have representations that are more similar to models trained with a predictive easy feature. In fact, the multi-task model, which is trained to classify *both* features, produces representations that closely resemble those of the model trained with a highly-predictive easy feature. This suggests that the easier feature dominates the representations of the multi-task model, even when it also represents the more difficult feature. This finding suggests that there are cases in which the value of a representational similarity metric may be dominated by the representational structures induced by one feature, for example an easier one, even if another feature is still represented.

Figure 8: **Representational similarity of Binary Feature models.** We compare an untrained model, a model trained with an unpredictive, or perfectly predictive, easy feature ($p(\text{label}|\text{easy feature}) = 0.5$ or $1.0$), and a multitask model. While similarity is generally highest between pairs of models trained on the same task, the numerical magnitude varies considerably. Multi-task models closely resemble the model trained with the predictive easier feature, even though they are also trained to output the more difficult feature. (5 runs per condition.)

## 6 Conclusion

Overall, our results sketch a picture of how a model's representations are shaped by its inductive biases and training. In the presence of multiple redundantly predictive features, the model may choose to principally enhance one, and this choice will favor the feature that is most decodable from an untrained model. Indeed, the model will sometimes favor features that are easier in this sense over ones that are harder, even if the latter are more predictive of the label. Furthermore, features that are not used for classification are actively suppressed rather than preserved from the initial state, even in some cases when they perfectly predict the label. Still, these features are not completely compressed away – in many cases they are still partially decodable. Finally, representational similarity is dominated by easier features, meaning that it may be misleading in complex or multi-task settings. We have shown these results across multiple architectures and datasets, but future work should explore broader settings.

Our findings suggest that, first, practitioners should not assume that label-relevant features will be used, or even represented by, a model. This may help explain shortcut learning, and the variable benefits of pretraining, since features useful for downstream tasks may be suppressed by pretraining. Second, analyzing untrained model representations can be a cheap way to predict which features a model might use. Finally, representational similarity analyses should be interpreted with care when applied to a model (or brain) trained on multiple features or multiple tasks.

## Broader Impact

Understanding the representations that models use to make their decisions is an important part of ensuring the correctness and safety of these decisions, particularly in situations where models operate on inputs different in distribution from those on which they were trained. Furthermore, many ethical concerns about machine learning models hinge on their reliance on features that encourage bias on socially discriminatory features; understanding and being able to predict which features are used is an important step in solving this problem.

## Acknowledgments and Disclosure of Funding

We thank Jay McClelland for useful conversations. KLH was supported by NSF GRFP grant DGE-1656518. AKL was supported by NSF GRFP grant DGE-114747.

## Footnotes

[2]Not enforcing feature correlations for the validation set would also be reasonable, but our approach matches the many real-world datasets where train and validation data are sampled from the same distribution.

[3]For reasons of computational resources, for ResNet-50 models, we sampled conditional probabilities as [0.1, 0.3, 0.5, 0.7, 0.8, 0.9].

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
