[Supplementary Material]

# Supplementary Material for
# "What shapes feature representations?
# Exploring datasets, architectures, and training"

## A  Supplemental Figures

Figure A.1: **Feature decodability in models with a ResNet-50 architecture trained on the Navon dataset.** Accuracy decoding features (shape, texture) from an untrained model (left) versus from shape- (center) and texture-trained (right) models. Results corresponding to trained models are mean across models trained on 5 cv splits. Chance = $\frac{1}{23}$ = 4.3% (dashed line). Target features are enhanced relative to the untrained model, whereas non-target features are suppressed.

Figure A.2: **Non-target features are suppressed in the post-pool layer of models with a ResNet-50 architecture trained on the Trifeature dataset.** Accuracy decoding features (shape, texture, color) for models trained to classify shape (left), texture (center), or color (right). Chance = $\frac{1}{7}$ = 14.3%. As observed with the Navon dataset and in the AlexNet models, non-target features are suppressed in trained models relative to the untrained model.

Figure A.3: **Non-target features that are correlated with the target feature are suppressed in ResNet-50.** For both datasets in which shape and color (upper row) and shape and texture (lower row) are correlated, target features are enhanced (left column), whereas non-target features correlated with the the target feature ("correlated non-target feature", right column) are suppressed. As observed in experiments using an AlexNet architecture, suppression of the correlated non-target feature is largely constant across correlation strengths (x axis).

Figure A.4: **Decoding accuracy for models with an AlexNet architecture trained on the Trifeature (Correlated) datasets.** See Figure 5 for these results plotted in terms of enhancement/suppression relative to an untrained model.

Figure A.5: **Decoding accuracy for models with a ResNet-50 architecture trained on the Trifeature (Correlated) datasets.** as a function of the degree of correlation of the non-target feature with the target feature in See Figure A.3 for these results plotted in terms of enhancement/suppression relative to an untrained model.

Figure A.6: **When two features redundantly predict the label, models with a ResNet-50 architecture preferentially learn one feature.** Color is more decodable than shape, and shape is more decodable than texture, when ResNet-50 is trained on perfectly predictive pairs, consistent with the pattern we observed in AlexNet (Figure 4).

Figure A.7: **Models with an AlexNet architecture sometimes suppress features that perfectly predict the label.** Models trained on a dataset in which shape and color redundantly predict the label suppress shape in pool3 and fc6 relative to an untrained model. Models trained on a dataset in which shape and texture predict the label suppress texture in pool3.

Figure A.8: **Models with a ResNet-50 architecture sometimes suppress features that perfectly predict the label.** Models trained on a dataset in which shape and color redundantly predict the label suppress shape in the post-pool layer relative to an untrained model. For reasons of computational resources, we did not decode from the pre-pool layer.

## A.1 Binary feature tasks

Figure A.9: The dynamics of feature learning when the easy feature has relatively low predictivity (0.65) and the hard feature has high predictivity (0.9). The easy feature is decodable above chance ($\sim 75\%$, chance is 50%) before training (epoch 0), while the difficult feature is not. Perhaps because of this, the easy feature is learned first, and test performance on this feature, as well as its decodability, spike early. As the more difficult feature is learned, the decodability and use of the easier feature declines, although not to chance, and the more difficult and predictive feature is still decodable and used less than 80% of the time. (Averages across 5 runs.)

(a) Nonlinear decoder accuracy.

(b) Advantage over linear decoder.

Figure A.10: Nonlinear decoders trained on a larger dataset (2048 examples) are still unable to recover the more difficult feature when it is suppressed by the easier feature. (a) The nonlinear decoding accuracy of the difficult feature still declines drastically when the easy feature is more predictive. (b) While the nonlinear decoders do have some advantage over linear decoders (trained with the same 2048 examples), particularly when the easy feature has moderately high predictivity, the magnitude of the advantage is small. (Results are from same models reported in Fig. 6, with only the decoders trained on the larger dataset. The nonlinear decoders were a 2-layer fully connected network, with 64 hidden units. Panel a contains a linear model fit, while panel b contains a loess curve.)

## A.2 RSA

(a) Similarity by architecture and task pairing.

(b) Details of cross-task similarity.

Figure A.11: Two views of the RSA results on the Navon tasks. (a) The results across the two architectures we considered, and different possible training task pairings: two different training tasks, an untrained model vs. a trained one, and two models trained on the same task. In general, whether models are trained on the same task significantly drives RSA results, although untrained models do have some similarity to trained models. (b) Representational similarity between all models and layers on each possible pair of tasks. While similarity is highest between models trained on the same task, the magnitude of that similarity varies across tasks. For instance, two texture-trained models are much less similar to one another than shape trained models. See Fig. 7 for equivalent analyses on the trifeature tasks. The overall results are similar, but the Navon dataset shows slightly higher similarity within models than the trifeature dataset, and slightly more of an effect of architecture on representational similarity. (Results are from 5 runs per condition.)

Figure A.12: The effect of feature correlations (between shape and texture) on representational similarity in the trifeature dataset. The representational similarity analysis is most sensitive to the target feature, even when another feature is relatively strongly correlated with it. (Results from 5 runs per condition, with matched architecture and layer.)

(a) RSA with correlation distance similarity.

(b) RSA with Euclidean similarity.

(c) CKA.

Figure A.13: Further representation analyses for the toy tasks. Here we expand the results from the main text figure Fig. 8, including both comparisons to intermediate easy feature predictivity values, and different analysis approaches. The results for intermediate easy feature predictivities interpolate between the cases shown in the main text, but this interpolation reflects the bias towards the easier features. In the main text (Fig. 8) we used RSA with correlation distance as the metric. We show an expanded version of this figure in a). We also show that (b) RSA with Euclidean distance as the metric, and (c) CKA [22] both yield quite similar patterns of results. Our conclusions do not appear to be limited to the particular analysis we considered in the main text.

Figure A.14: An intuitive example of why representational similarity might be lower on nonlinear tasks — there are multiple solutions resulting in different RDMs. We consider two possible ways that a network with two hidden units could compute the XOR of two binary inputs: either by an AND and an OR (top row), or by units that select each of the valid combinations (bottom row). When the inputs are passed through these networks, and representations are computed at the hidden layer, the patterns are different. In fact, when the representational dissimilarity matrices are computed, and then the correlation is taken between the two networks, it is actually negative! This raises the question of why positive representational similarities are observed at all between different networks computing XOR. The answer likely lies in overparameterization — a very large hidden layer will likely contain units which partially represent both solutions. However, the network will still likely favor one or the other, thus resulting in lower correlations between RDMs than on simpler tasks. (This will likely be exacerbated by other features being partially represented, and so on.)

# B    Methods

## B.1    Decoding

### B.1.1    Decoding from vision models (Sections 3 and 4)

**Layer definitions**. For AlexNet, we decoded from "pool3" (the output of the final convolutional layer, including the max pool), "fc6" (the first linear layer of the classifier, including the ReLU), or "fc7" (the second linear layer of the classifier, incluing the ReLU). For ResNet-50, we decoded from "pre-pool" (the output of the final convolutional layer prior to the global average pool) and "post-pool" (after the global average pool).

**Training procedure**. The inputs to a decoder were activations from some layer of a trained, frozen model in response to images normalized by the statistics of the dataset on which the model had been trained (images were unnormalized when decoding from untrained models). We trained and validated decoders on either a version of the Trifeature dataset (when decoding from models trained on a Trifeature task) or a version of the Navon dataset; in both, sets of features were uncorrelated.

We trained decoders to minimize cross-entropy loss for 250 epochs using Adam optimization [20] and a batch size of 64 for each of 6 learning rates $(10^{-6}, 10^{-4}, 10^{-3}, 10^{-2}, 10^{-1}, 1)$ and 6 weight decays $(0, 10^{-6}, 10^{-4}, 10^{-3}, 10^{-2}, 10^{-1})$. We selected the decoder with the highest validation accuracy (when using trained models, we had trained models on 5 cv-splits, so took the mean validation accuracy across the 5 trained models) over the training period over the set of hyperparameter combinations.

### B.1.2    Decoding from models trained on binary features (Section 4.3)

Decoders were trained for 5000 epochs with a learning rate of $10^{-3}$. We did not use weight decay or search over hyperparameters for the binary feature datasets. For the nonlinear decoding analysis, we used decoders with a single hidden layer with 64 units and a leaky-rectifier nonlinearity, trained for 20000 epochs.

## B.2    Datasets

## B.3    Trifeature

In Fig. B.1 we show sample stimuli illustrating the range of features in the trifeature dataset.

Figure B.1: Sample images showing all colors, shapes, and textures used in the trifeature dataset.

### B.3.1    Binary features

As described in the main text, the datasets were composed of inputs that were 32-element binary vectors, and outputs (labels) that were single binary scalars (except in the multi-task case). We divided the inputs into two domains of 16 inputs each, and the labels were probabilistically related to

a feature extracted from each domain. Each feature had a predictivity ($p$ (label|feature)). We sampled the datasets by first assigning a label of 1 to half of the inputs, and then for each domain flipping the label with probability $1 -$ predictivity, and then sampling a domain input uniformly from the set of inputs that matched that feature value. For example, if the label was 1, and we flipped the label for the XOR feature, we would sample an input where the first two inputs of the XOR domain had parity 0.

## B.4 Representational Similarity Analyses

For the Trifeature datasets, representational similarity analyses were performed using 10 examples per combination of features of the 100,000 uncorrelated images. These images would sometimes have some small overlap with the train and val sets, but it was difficult to exclude the sets for all images, and generally due to the small sizes of the train sets this overlap would be no more than a 10% of the RSA dataset. Similarly, for the Navon dataset, we performed the representational similarity analyses on all images. We found that RSA on only train or only validation data did not produce very different results between Navon models for which it was easy to make the comparison.

For the binary feature datasets, representational similarity analyses were performed on a new dataset sampled independently from the training and evaluation data for the models; the predictivity of the features in this dataset did not matter, since the labels are not used in RSA, only the patterns of activation produced by the inputs.

We also compared to CKA [22], as well as RSA with Euclidean distance as the RDM metric, and found similar results in both cases (Fig. A.13), and similarly with Spearman rank correlation rather than Pearson between the RDMs.