[Reviews · NeurIPS 2020]

Review 1

Summary and Contributions: The paper explores how deep neural model see the data while solving a task-specific learning problem. For this purpose, the approach considers AlexNet and ResNet50 architecture to understand high-level feature representation and their usefulness for the task in hand. In order to understand this, the approach use synthetic datasets of Trifeature and Navon so that the input image can be controlled to measure the respective outputs. The paper tries to answer questions of model’s decision-making process by exploring feature types (shape, color, and texture), their correlation and the task in hand. The experiment also involves understanding the consistency in feature representation across model, as well as the model’s bias towards easy to difficult features. The results also highlight the process that determine which feature a model represents.

Strengths: A novel framework to understand the process involved in deep models during decision-making. This work is relevant to NIPS/AI/Computer Vision communities. It gives insight into a deep models’ decision-making process. The framework explores the three primitive features (shape, color, and textures) and analyses how a model’s representations are influenced by its inductive biases and training. In order to achieve this, the framework uses ResNet50 and AlexNet architectures. These architectures are adapted to experiment with synthetically generated datasets sampled from Navon and Trifeature dataset. The experiments are design to measure various input-output relationships. This is carried out by controlling inputs and measuring the corresponding outputs. The experimental evaluations provide insight into how feature representations are affected by learning and how to analyse them.

Weaknesses: One of the main concerns is the size of the dataset used from training (4900 images). To train a deep architecture, this size is very small. It is well-known that deep models using smaller datasets often result in a lower test accuracy, perhaps because the training set is not sufficiently representative of the problem and the model might overfit. To address this, researchers have of the used transfer learning (pre-trained base CNNs that are trained over the large diverse datasets) and fine-tuned on the smaller dataset. Given the size of the AlexNet (61M parameters), I have a feeling that the model is overfitted for this particular experimental design and evaluation. In the untrained models, are model initialized with random weights or taken from the pre-trained model (e.g. ImageNet). This has not been clarified. AlexNet is one of the first CNN architectures. Since 2012 there has been significant advancement in CNN architectures (e.g. Inception, Xception, DenseNet, etc.) and thus, it would have been beneficial if the experimental evaluation has been carried out using one of these latest architectures. During the training of decoding experiments, the base models (ResNet and AlexNet) were frozen and these base models are trained on a different set of labels. This makes two doubts: 1) why base models trained on different set of labels when the goal is to measure the influence of the type of features (shape, color, and textures) and 2) if layers in base models are frozen during decoding than why not use the base model trained on natural images (e.g. ImageNet) The selection of upper layers in the base models. Is there any rationale behind the selection of pool3 layer? In pooling layers, there are no learnable parameters associated with them and thus, one would expect features extraction from other types layers (e.g. CONV and FC). It would have been appropriate to select features from last CONV layer. The reduction in number of units in fc6 and fc7 layers is not justified. In the original model, a significant contribution towards the performance is from these two layers. To my view, the reduction in units in these two layers has impacted the performance of the model. What are the target classes in both datasets (Navon: 23, Trifeature: 7)? Is it a unique combination of shape, color, and texture? Or their rotational and positional combinations as well. How the samples are generated using correlation? It is very important for predicting multiple features, but the explanation is unclear. For example, how do you generate image with conditional probability. For example, if one wishes to generate a sample shape to match color of 0.3. Is this implying that 30% of the given shape has the target color? If yes, then how that 30% of the shape is chosen? One chunk of 30% or randomly over entire shape? This is unclear. Moreover, it is also unclear what color is given to the rest 70% of the shape. The binary features datasets are unclear as well. What is 32-element binary vectors and how it is selected from the input dataset. How do you define easy feature and difficult features?

Correctness: The proposed methodology seems to be correct. The empirical methodology also seems to be correct. However, there are drawbacks in experimental design and explanations. This has been mentioned in the section “Weakness”.

Clarity: The paper is well-written and easy to follow. However, it could have been improved with a bit more details about the creation of correlated and binary features. This has been pointed out in section “Weakness”. The additional supplementary information also provides valuable support for the experimental analysis.

Relation to Prior Work: The paper discusses about the important related work. A brief explanation of the key message in the related work is provided. However, there is no relationships/explanation on how the proposed approach differs from these. It would have been beneficial to highlight the drawbacks in the existing approaches/experiments and how the proposed experimental design and evaluation address them.

Reproducibility: No

Additional Feedback: It is a good piece of work to get insight into the deep models. However, there is lack of clarity in many areas (please refer to the “Weakness” section), as well as rationality in experimental design, choice of architectures, and the selection of dataset size. This has significantly impacted the overall rating of this work. I have given major comments in the “Weakness” section and please refer to it. Minor issue: The pdf files are unable to open in Adobe Reader. It gave error ("The document's page tree contains an invalid node") but I managed to open with browser. *********** Post-Rebuttal Update ************* I would like to thank authors for their response. I have gone through the rebuttal and it addresses some of my concerns. I have also gone through the reviews of other reviewers. One of my concerns which is "generalizability" (also raised by R4) of the model experimented using synthetic datasets in comparison to realistic settings. This is not addressed in the rebuttal. By considering all these information, and some novelty in the experimental design and evaluation, I would stick to my original rating which is 6.


Review 2

Summary and Contributions: ========Update after rebuttal============ I've read the author rebuttal and the other reviews. I am generally satisfied with the author's rebuttal, which managed to address most of our concerns. Regarding decoding from earlier layers, I think it would indeed be valuable to include earlier layers to see how the "suppression" and "enhancement" occurs over the course of the whole network depth. The primary remaining concerns, raised by other reviewers, concern the difficulty of the features and whether the results of these toy problems would generalize to more naturalistic settings. Regarding the first concern, I do not find this to be as large of an issue as reviewer 3. However, I can see that some experiments would have benefitted from a condition where the difficulty was held constant. For example, the question of what a neural network learns when two features redundantly predict the target. For completeness, it would be good to show what happens when the two features are equally "easy" to learn while varying their predictiveness and correlation to better understand these various factors and how they interact. Regarding the second concern about generalizability. Highly controlled experiments like this are not possible in naturalistic settings, but I think they are valuable none the less. The authors have promised to add more discussion of related work, including work on natural images, which may help to bridge the gap. I maintain that this is a good paper that should be accepted. The paper will be stronger if the authors take into account reviewer feedback in the camera-ready. ==================================== A number of synthesized datasets are used to investigate questions about how task-relevant and task-irrelevant information is transformed and what features of the input data are used to make predictions under various training scenarios. They report a long list of interesting experimental findings. setting: all features are independent, only one feature is predictive findng: task-relevant features become more decodable after training. Task-irrelevant features generally become less decodable after training. setting: two features redundantly predict the target finding: rather than learning both, models privilege one feature over the other in a way that reflects the linear decodability of the features from the activations of the model before training. setting: one feature perfectly predicts the target while another only partially predicts it finding: networks are sometimes "lazy" meaning they'll "suppress" a more predictive feature in favor of a feature that is more linearly decodable at the beginning of training.

Strengths: * The results in this paper are important for understanding phenomena like shortcut learning where networks find unanticipated, simplified solutions to difficult cognitive problems that only work in narrow settings. In general, the paper contributes to understanding what deep networks learn, a topic of interest for many in the NeurIPS community. * The paper presents well controlled experiments that allow the authors to precisely adjust the learning problem, e.g. modulating how correlated the features are * The experiments are repeated in two architectures and 5 partitions of the data.

Weaknesses: * The analysis only considers three layers (pool3, fc6, fc7). It would be informative to see all layers. * The submission never discusses the difference in spatial frequency between the different features. Information related to texture will be encoded in higher spatial frequencies than shape and both shape and texture will be present at spatial frequencies higher than color (i.e. you could low-pass filter the images and still classify color while the other features would be more affected.) I don't think this changes the interpretation of the results, but it likely partly explains the specific patterns that are observed and so would be good to mention. * Only empirical observations: there is little attempt to relate the results to existing DL theory. For example, Andrew Saxe talks about how (depending on the nonlinearity) networks learn linear functions first, and then, later in training, take advantage of their nonlinearities to learn more complex functions (c.f. his DL Indaba talk). This could help to explain the "lazy" models.

Correctness: The empirical methodology is correct to the best of my knowledge. I was a bit worried about the number of training examples being low and the architectures used being overkill for these tasks. I'm not sure why the authors didn't use smaller networks, which probably would have made their experiments much quicker.

Clarity: * I don't have any major issues with the quality of the writing but there are a few aspects of the description that I think might be confusing for a NeurIPS audience. e.g. the text uses 'feature' and "input feature" to refer to parameters of the data generating process when the pixels of the images are the input features to the networks being probed. These 'features' also become 'labels' sometimes, depending on the task. In deep learning we typically refer to each layer as learning a feature and so it is not immediately obvious what is meant by this word here until you read through most of the paper. * The submission uses a number of vague terms and expressions. For example, training on a task "enhances" some features while "suppressing" others, or describing features as being "easy", "hard" or models as "lazy". There are a variety of possible precise meanings of these terms so the submission would benefit from being very clear from the beginning about what is meant by these terms. In general, easy to express is not necessarily easy to learn is not necessarily linear. The related work section mentions the information bottleneck theory which primes me to think we're going to be talking in information theoretic terms but "easy" here seems to be operationalized by how linearly decodable a feature is in an untrained network. Minor issues * line 142-144: sentence beginning Non-target features: you mean "first" not "final", right?

Relation to Prior Work: Prior work is clearly reviewed.

Reproducibility: Yes

Additional Feedback: I don't think it is needed for this paper, but it would be interesting to see how all of these results develop over the course of training.


Review 3

Summary and Contributions: ====update==== After discussing with other reviewers and walking through the authors' response, I still find it difficult for me to raise my score to anywhere close to accepting this paper. The very first step of experimental design didn't control for the task difficulty, and all experimental results could easily be an artefact of that and they couldn't tell us anything about feature representations and learnability of neural networks. My concern on publishing this paper also comes from the potential influence on other researchers in the field that they might consider the task design was legit, whilst it was not well-controlled at all. ====end of the update==== The paper presented several tasks to evaluate what features are learnt, and maintained and suppressed in neural networks. With selected three low-level vision tasks, including colour, shape and texture recognition, the authors made interesting findings that neural networks generally prefer simple features even if hard features contain more predictive information about the labels.

Strengths: The paper has a clear presentation of the design of datasets, and the analysis of results.

Weaknesses: ((1)) I hope that the authors have a grasp of manually designed image features and their application in the computer vision research area before the deep learning era. That's being said, I don't think the proposed three aspects/features are reasonable as classification tasks defined by them, respectively, are at the same difficulty level. A model can categorise various colours by simply looking at pixel values, which is the simplest task among three presented ones. A slightly more involved feature engineering is required for shape recognition as either the designed feature or the classification model needs to be invariant to affine transformation, which makes the task itself harder than categorising colours. The hardest task among three is texture recognition as it requires proper statistical features or Fourier features of individual texture patterns to solve the task. Therefore, given the understanding of the task difficulty provided by classic CV research area, results presented in the paper are all predictable and expected. For example, in Fig 2, texture classification task always has the lowest accuracy on an untrained model, on a model trained on non-texture tasks. in Fig 4, when a model is learnt on correlated features, the model always picks the easiest one to learn from as it makes the model simple. ((2)) From a theoretical point of view, machine learning models tend to learn simple functions rather than difficult ones, so that when tasks with different levels of difficulty are presented, models tend to first learn to do well on simple tasks. Recent work on randomly permuting labels show that machine learning models first learn from easy samples then move to hard samples, which could be used to explain the observation here as well without conducting experiments. Some example work can be found from Belkin's group. ((3)) There is a transition happening in NLP area that people started using nonlinear classifiers to 'decode' information from pretrained language models for downstream tasks. People tend to use linear classifiers as the argument was to check whether the learnt representations were general enough so that it could be 'decoded' linearly. However, as for making predictions for a new task, nonlinear ones are preferred as those classifiers can 'abstract' information that is relevant to the task. Therefore, here, I think nonlinear ones are more appropriate. ((4)) The defined correlation along with some results presented in Fig. 5 works better with regression tasks rather than classification one as the correlation is related to error measures in regression tasks. Therefore, I would recommend the authors consider regression tasks for decoding. ((5)) RSA is the Centralised Kernel Alignment with linear kernels. From this perspective, prior work (Ref [20]) has already done extensive experiments on comparing the similarity of neural networks using CKA with linear kernels. I recommend the authors to check the equivalency between RSA using Spearman Rank Correlation and CKA with a linear kernel.

Correctness: Yes, apart from some concerns in the task design, the paper has done a decent job.

Clarity: Yes, the paper is well-written and has made a clear delivery of the experimental design and results.

Relation to Prior Work: The paper is missing a large body of work in the classic Computer Vision area on solving colour, shape and texture recognition, and also missing recent works on comparing the similarity of neural networks and transferability.

Reproducibility: Yes

Additional Feedback: Let's look into all figures, and the results can all be determined or explained by the task difficulty without any other additional design on the correlation between features and labels. Reviewers agreed on that texture recognition is harder than shape recognition, which is harder than colour differentiation, so let's start from there. Fig 2 (a), on the untrained model, the accuracy of colour is higher than that of shape, which is higher than that of texture. Fig. 2(b), on a model trained with shape recognition, shape has the highest accuracy and texture has a lower accuracy than colour does. Fig. 2(c), on a model trained with texture recognition, texture has the highest accuracy and shape has a lower accuracy than colour does. Fig. 2(d) says a similar thing. Fig. 3: it represents that the model trained on one task, including shape and texture, doesn't transfer well to the other. Fig. 4: indeed as we agreed on, as colour is the easiest task among all three, it is always easy to decode the colour task than the shape task, and the texture one is the hardest one to decode. The plots can be predicted directly from the task difficulty itself regardless of the choice of model. One can reproduce their results simply with a simple classifier with bag-of-features representations. How does it help us understand what neural networks have learnt or how neural networks behave when "two features redundantly predict the label"? What additional information does this plot provide besides predictable results simply given the task difficulty? Fig. 5: The difficulty level of individual tasks explains the observation that the suppression level of correlated non-target features is consistent across different correlation values. My point here is the same that the results are predictable purely given the task difficulty level, so then we still don't know what neural networks would behave when we have different tasks at the same difficulty level but with varying correlation values. Fig. 7: the feature similarity has been extensively studied within the neurips community, and recent work from Kornblith et al. ICML2019 conducted a comprehensive study on the proper measure for comparing the similarity between neural network features. The issue I have here is that Fig 7(b) is also predictable given only the task difficulty. Since colour is the easiest among all three tasks, a trained model on this task won't deviate much from an untrained model, therefore the untrained model has highest similarity with the colour task compared to others.


Review 4

Summary and Contributions: This paper investigates feature representations in trained and untrained deep nets using toy classification tasks. It finds that features are decodable even from untrained models. Task-relevant features are amplified and task-irrelevant features are partly suppressed in trained models. When multiple features are predictive of class, the trained model picks one of the features (the one most dominant in the untrained model). Models seem to prefer color over shape, and shape over texture. The paper also finds that sometimes less reliable but easy-to-learn features can be preferred by the model over more reliable but more difficult-to-learn features. Models trained on the same task have similar representations. The effects of other factors on representational similarity are rather small.

Strengths: I think the questions addressed in this paper are important and of general relevance to the machine learning community.

Weaknesses: The main weakness of the paper is that it relies exclusively on toy tasks. This is, to a certain extent, understandable when one wants to run controlled experiments, but for me it raised some concerns about the generalizability of the results at times. For example, the authors find that their models prefer color over shape, and shape over texture. The latter is a bit concerning since it seems to contradict recent results suggesting that deep convnet preferentially learn textures over shapes in more realistic settings (Geirhos et al., ICLR 2019; Brendel & Bethge, ICLR 2019). This suggests that the tasks in this paper may perhaps be a bit too simplistic and unrealistic. Other than this, I don’t have any major concerns about the paper.

Correctness: Yes, I found the experiments quite rigorous and informative.

Clarity: Yes, the paper is generally very well-written.

Relation to Prior Work: Yes, generally speaking, prior work is adequately discussed to the best of my knowledge. However, I would just suggest that the authors should perhaps acknowledge the contributions of prior works more generously in the related works and conclusion sections (the conclusion section in particular doesn’t have any citations). Several of the findings in this paper had been explicitly predicted or anticipated in prior work: for example, the finding that when multiple features are predictive of class, the trained model picks the “easier” feature (Geirhos et al., ICLR 2019; Brendel & Bethge, ICLR 2019; see especially section 6.3 in Geirhos et al., 2020 [ref. 8]).

Reproducibility: Yes

Additional Feedback: ====== Update after rebuttal ====== I have read the rebuttal and the other reviews and decided to keep my score the same. I did not have any major concerns about the paper in the initial round. During the discussion period, another review raised an objection regarding the possibility of task difficulty (i.e. difficulty of learning different features) being a confounding variable. However, I do not share this concern as the authors are explicitly considering ease of learning as a factor that might influence what the model learns. So it seems OK to me, in fact necessary, to use features with different difficulty levels. ============================ Additional comment: - Figure 8 is a bit hard to follow. I wonder if there is a more intuitive way of representing the results in this figure: for example, a 4x4 heatmap (p=0.5, p=1.0, multitask, untrained) may perhaps work better; the intermediate points between 0.5 and 1 in this figure are clearly not essential.

[Author Response · NeurIPS 2020]

We appreciate that the reviewers found our work "important" (**R2** & **R4**), "interesting" (**R3**), "novel" and "relevant
to the NeurIPS/CV/AI communities" (**R1**); found our experiments "well-controlled" (**R2**) and "quite rigorous and
informative" (**R4**); and found our presentation "clear" (**R1**, **R3**, **R4**). We thank the reviewers for their helpful comments.

**Feature difficulty (R3):** *"I hope that the authors have a grasp of manually designed image features and their
application in the computer vision research area before the deep learning era...Results presented in the paper are all
predictable and expected."* We agree that color is an easier feature than shape or texture. However, the goal of our work
was **not** to reason about the absolute difficulty of *particular* features (shape, color, texture). Rather, we were testing
what deep models come to represent when multiple features are present and correlated to differing degrees with task
labels, regardless of what those features are. We performed experiments using both vision and non-vision datasets.
Indeed, we found that feature difficulty was not the sole determinant of feature use or representation (Figs. 5 & 6).

**Decoding experiments (R1):** We tested which features a model represents as a function of training task, which required
using a carefully controlled dataset. The joint image feature–label statistics of ImageNet are unknown and uncontrolled.
We see our work as complementary to existing work that has looked at feature representations and feature-based
classification behavior in ImageNet models (e.g. Geirhos et al. 2019, Brendel and Bethge 2019, Hermann et al. 2019).

**Choice of vision model architectures**. *Probe a post-AlexNet architecture* (**R1**): We also performed our vision
experiments in ResNet-50, a standard benchmark in computer vision (see Appendix A). *Concern that AlexNet would
overfit a small dataset* (**R1**): It did not in our case. Our validation sets require generalization over held-out features (see
Sec. 3). *Choice to reduce FC layer widths* (**R1**): It is standard practice to reduce classifier sizes in proportion to the
number of output classes (see e.g. Qian et al. 2020). Nonetheless, we have done initial experiments with standard FC
widths, which show somewhat worse validation performance for texture (and comparable performance for shape and
color) compared to the models reported in the paper, when using our current set of hyperparameters.

**Connections to the literature.** *Theory on learning simple to complex* (**R2**, **R3**): The theoretical results of Saxe, Belkin,
and others are relevant. However, that work leaves open questions of the effect of multiple redundant features with
varying predictivity, which our work attempts to address. We show aspects of feature learning dynamics over training
(Supp. Fig. A.9) that would not be completely predicted by Saxe's theory. We will discuss how our work complements
prior work in the revision. *Spatial frequency* (**R2**): We agree that spatial frequency is related to the features we
investigate, though we don't see it as equivalent. We will add citations to recent related work (e.g. Yin et al. 2019).
*Further discussion of related work on shortcuts, etc.* (**R4**): We will add further discussion of these papers, which we see
as complementary and related. We note that Geirhos et al. investigated classification behavior (not representations), and
that dataset statistics influence whether texture bias is observed (see Geirhos et al. 2019, Hermann et al. 2019).

**Correlated trifeature datasets (R1, R3):** A pair of features (e.g. shape and color) was correlated across the set of
images (not within individual images). As an example, suppose that shape and color are correlated with conditional
probability = 0.5. Then, in images containing triangles, half would be red and half would be uniformly sampled from
the other colors (blue, white, etc.). Similarly, in images containing trapezoids, half would be orange and half would be
some other color. The attribute matching (e.g. triangle = red, trapezoid = orange, etc.) was randomly chosen. Although
regression decoding is interesting (**R3**), our experiments also contribute, since classification is a common task.

**Binary features datasets (R1):** One measure of task difficulty is whether the task is solvable by a single layer, or
requires a multi-layer perceptron to solve it, e.g. XOR (Minsky & Papert, 1969). In our (non-vision) Binary Features
dataset, we defined two features: one that is learnable by a linear model ("linear" feature, which we call "easy"), and
one that requires an MLP (XOR, a "nonlinear" feature, which we call "difficult"). The inputs for this dataset were
32-element binary (1 or 0) vectors in which the first 16 elements instantiated feature A (linear), and the second 16
instantiated feature B (nonlinear). The label was probabilistically determined by these two features — that is, the
inputs were sampled so that each feature matched the label a certain percentage of the time. To probe the model's
computations, we created new datasets where e.g. the label matched feature B and was uncorrelated with feature A.

**Nonlinear decoders (R3):** We found similar results on binary features with nonlinear decoders (Supp. Fig. A.10).

**Decode from additional model layers** (**R1** & **R2**): We considered the output of the convolutional layers because this
high-level visual representation is the standard choice for transfer to downstream tasks, and classification layers because
these determine the model's classification decision. We agree that it would be interesting to additionally consider earlier
model layers in future; we are happy to do so before the camera-ready if the reviewers feel this would be beneficial.

**Terminology (R2):** We will clarify in the introduction that our "features" are latent variables within the data-generating
process underlying the inputs, not necessarily accessible at the pixel level, and that uncovering when and where these
latent variables are represented in the model is our goal. We provide some clarification of our definition of difficulty
above ("Binary Features Datasets"). We will move our definitions of "suppression" and "enhancement" earlier in text.

**RSA (R3):** Prior work doesn't perform RSA on models trained on different features. We compare to CKA (Fig. A.13).

[Meta-Review · NeurIPS 2020]

The paper studies how task-relevant and task-irrelevant features are used to make predictions on artificial datasets (Navon, Trifeature) where the relationship between input features (color, shape, texture) and labels can be controlled for. The paper has a few interesting insights that are useful to understanding deep nets (e.g. when multiple features predict the label, models prefer easily-decodable features). Two main concerns raised by the reviewers (and left unaddressed in this work) are (1) whether the results here transfer to the real-world settings e.g. ImageNet and (2) the difficulty of features is not controlled. Overall, this is a good paper but definitely with rooms for improvement.